# Experiences and Lessons Learned from Designing and Testing of an Air System and a Drilling Fluid Circulation System Adapted for Subglacial Bedrock Sampling in Antarctica

Yazhou Li<sup>1, 2</sup>, Gansheng Yang<sup>1, 2</sup>, Jing Wang<sup>1</sup>, Kai Zhang<sup>1</sup>, Bing Li<sup>1, 2</sup>, Yangxi Lu<sup>3</sup>, Zheng Zhou<sup>1</sup>, Zufang Wang<sup>1</sup>, Ruozhou Huang<sup>1</sup>, Xingwen Lai<sup>1</sup>, Yuchen Sun<sup>1</sup>, Mingqi Wang<sup>1</sup>

10 Correspondence to: Kai Zhang (zhangkai66@cugb.edu.cn)

Abstract. Liquid drilling is commonly utilized in sampling of subglacial bedrock in Antarctica. However, this drilling method has relatively low penetration rate compared with air drilling. Additionally, the drilling method may lead to hydraulic fracturing of ice borehole. In this study, a multi-process drilling system (MPDS) incorporated with different drilling methods, comprising an air system and a drilling-fluid circulation system (DFCS), was developed for sampling the subglacial bedrock in Antarctica. The air system uses a compressor to generate compressed air at a flow rate of 10 Nm³·min⁻¹ and maximum pressure of 1.5 MPa. The compressed air was then dried by a freezing dryer and a desiccant dryer to a dew point of -40℃. Before injected into the borehole, the compressed air was cooled to ≤-5 °C by an air cooler. The DFCS can pump drilling fluid to the borehole at a flow rate of 100 L·min⁻² and maximum pressure of 2 MPa. The drilling liquid can be cooled to ≤-5 °C by a refrigerating machine and a heat exchanger within DFCS. The ice or rock cuttings are separated by a vibration sieve and a vertical centrifuge. The two systems were integrated into modified 20 ft containers for easy transportation and assembly. Both systems worked with limitations: the failure of the freezing dryer and desiccant dryer led to the breakage of the air cooler in the field, and several problems were found in the drilling-fluid pump, vibration sieve, vertical centrifuge, and circulation tank. This paper presents in detail, the requirements, principles, and design of the air system and DFCS, in addition to the domestic and Antarctic test results. The experiences and learnings gained in this study will contribute to the development of ice and subglacial bedrock drilling technology.

#### 1 Introduction

Antarctica is widely covered by ice sheets, ice caps, ice shelves, sea ice, and glaciers. Approximately 2 % of the Antarctica land surface are ice-free (Singh et al., 2011). Subglacial bedrock, buried under tens to thousands of meters of Antarctic ice, is important for the study of the historical evolution of ice sheets, revealing geological tectonics and paleoclimate (Fountain et

<sup>&</sup>lt;sup>1</sup>School of Engineering and Technology, China University of Geosciences, Beijing, 100083, China

<sup>&</sup>lt;sup>2</sup>Key Laboratory of Polar Geology and Marine Mineral Resources (China University of Geosciences, Beijing), Ministry of Education, Beijing, 100083, China

<sup>&</sup>lt;sup>3</sup>Shaanxi Taihe Intelligent Drilling Co., Ltd, Xi'an, 710086, China

al., 1981; Schaefer et al., 2016; Bierman et al., 2024). In recent years, the science community has been very interested in obtaining subglacial bedrock samples from Antarctica (Talalay, 2014; Spector et al., 2018; Briner et al., 2022).

Many holes have been drilled in the ice-sheet beds in Greenland and Antarctica, but subglacial rocks have been rarely retrieved (Bentley and Koci, 2007; Talalay, 2013). There are two types of drills that have been used for subglacial bedrock drilling: conventional rotary drill rig and cable-suspended electromechanical drill.

Conventional rotary drill rigs with drill pipes have often served as the basis for subglacial bedrock coring systems. Examples include purpose-built adaptations like the Longyear Super 38 drill rig, the Winkie drill, the Agile Sub-Ice Geological (ASIG) drill, and the Rapid Access Ice Drill (RAID) (Truffer et al., 1999; Boeckmann et al., 2021; Kuhl et al., 2021; Goodge et al., 2021). The first subglacial bedrock core was drilled in 1956 with a conventional rotary drill rig near Mirny station, Antarctica. The hole was only 66.7 m in depth, and the rock core was 2.2 m in length (Treshnikov, 1960).

40 After that, the conventional rotary drill rig was abandoned for subglacial bedrock coring. Since 2010s, US researchers started to modify three conventional rotary drill rigs for subglacial bedrock drilling, and several subglacial bedrock cores have been sampled in West Antarctica and Transantarctic Mountains (Boeckmann et al., 2021; Kuhl et al., 2021; Goodge et al., 2021). Notably, RAID drilled the deepest subglacial bedrock borehole in Antarctica, reaching a depth of 681 meters, while the

The cable-suspended electromechanical drill can also be used in subglacial-bedrock drilling. For example, the Polar Ice Coring Office-5.2" (PICO-5.2"), Percussive Rapid Access Isotope Drill (P-RAID), and Ice and Bedrock Electromechanical Drill (IBED) were also designed to obtain the bedrock core beneath ice (Kelley et al., 1994; Timoney, et al., 2020; Talalay et al., 2021a; Talalay et al., 2021b). In last century, three subglacial bedrock cores had been drilled in Camp Century, Summit, and Taylor Dome by US scientists with cable-suspended electromechanical drill (Ueda and Garfield, 1968; Gow and Meese, 1996; Steig et al., 2000). In 2018/2019 season, Chinese drillers retrieved a short subglacial rock core at the margin of Princess Elizabeth Land, East Antarctica (Talalay et al., 2021a). Five years later, Chinese-Russian subglacial drilling project recovered a subglacial bedrock core of 0.48 m beneath 541 m ice sheet (Talalay et al., 2025).

longest subglacial core with length of 8 m were drilled by ASIG (Kuhl et al., 2021; Goodge et al., 2021).

Compared with the cable-suspended electromechanical drill, a conventional drill rig can drill longer rock cores. However, the aforementioned conventional drill rigs often utilize liquid drilling, which has relatively low penetration rate compared with air drilling and may lead to hydraulic fracturing of ice borehole. Air drilling has higher penetration rate in ice, however, the drilling depth is limited due to unbalanced ice pressure on the borehole wall. At present, the deepest hole drilled by air in ice is only 309 m (Patenaude et al., 1959; Lange, 1973).

Compressed air is generated by a compressor and is usually hot and wet. The compressed air must be cooled and dehumidified before being injected into an ice borehole. Otherwise, the ice borehole wall and ice cuttings may melt and condensed water may freeze on the borehole wall and drill pipes. For example, without cooling and dehumidification, the condensed water caused a stuck drill many times during air drilling in the Churlyenis ice cap, Franz Joseph Land, Russian Arctic (Bazanov, 1961). In past ice drilling projects, the compressed air was usually cooled by passive heat exchanger with

60

the atmosphere (Tongiorgi et al., 1962; Lange, 1973). However, this method is highly reliant on the ambient air temperature. For example, the Rapid Air Movement (RAM) drill used an aftercooler to cool compressed air, and it required ambient temperatures below ~10 °C, or the air will melt the ice chips (Bentley et al., 2009). Arenson (2002) proposed cooling compressed air in a 200 m long hose buried under the snow. However, this warm conduit could eventually melt the snow and freeze after circulation stops. Generally, the humidity of compressed air decreases during cooling, while there is still some water vapor left. To further separate water vapor from compressed air, RAM-2 drill use a coalescing filter (Gibson et al., 2021). Wang et al. (2017) suggested using both a coalescing filter and a desiccant dryer to dry the cooled air. In summary, the existing air-cooling techniques are insufficient to fully satisfy the requirements of drilling operations in Antarctic environments and new air-cooling method is required.

When liquid is used for drilling, cuttings must be removed from the drilling fluid. Winkie drill used simple filter bags to collect all particles larger than 100 µm (Boeckmann et al., 2021). ASIG drill and RAID used a commercially available screen shaker to separate ice or rock cuttings and the collected ice cuttings were melted in a heated melter tank to separate drilling liquid from melting water (Kuhl et al., 2021; Goodge and Severinghaus, 2016). Field test of the heated melter tank proved to be inefficient and power consumption. In the drilling-fluid circulation system (DFCS) of the Winkie drill, ASIG drill and RAID, there is no cooling system to cool the drilling liquid before pumping to the borehole. In low-elevation sites of Antarctica with warm ambient temperatures, the drilling liquid could become warm at the surface and degrade the drill performance, damage the borehole, and create a risk of equipment loss by freezing in the hole. Sometimes, the drill fluid drums had to be buried in the firn to cool it (Braddock et al., 2024). In addition, when the Winkie drill was used in subglacial bedrock drilling, the warm fluid near the bit melted the ice-borehole wall, and the melted water refroze in the casing, resulting in flakes that plugged the casing (Boeckmann et al., 2021). Therefore, a fluid chiller was suggested to cool drill fluid for ambient temperatures approaching -4 °C (Braddock et al., 2024). When drilling in ice, high pressure spikes can easily lead to hydraulic fracture (Chen et al., 2019). The use of pressure-relief valve at the outlet of drilling-fluid pump seems invaluable (Kuhl et al., 2021; Goodge et al., 2021). At present, the occurrence condition of ice hydrofracturing is still not clear and there is no effective way to prevent it from happening. In conclusion, the current method for separating ice chips requires improvement, and the integration of a cooling system into the DFCS is preferred.

To solve the above-mentioned problems of conventional drill rig, a conventional drill rig-based MPDS (Fig.1) incorporated with air drilling and liquid drilling, was developed in China for sampling at least 10 m bedrock beneath 1000 m ice sheet in Antarctica. The MPDS is designed to remove drill cuttings using compressed air or drilling fluid. The MPDS generally has five subsystems: drill rig, drill-pipe container, generator, air system and DFCS. All the subsystems are modularly designed and can be disassembled into several parts with each one less than 4 tons for easy transportation by helicopter from ice breaker to Antarctic ice sheet. Once arrived at ice sheet, all the subsystems can be integrated in a 20-foot container and can be transported on ice sheet by sledge. The drill rig is fully driven by a hydraulic system and can work with different drilling processes, such as air/drilling liquid reverse circulation drilling and wireline coring drilling. The drill-rod module is used for


drill rod storage and is adjacent to the drill rig for easy transferring of drill rod in field. Double-wall drill rod made by aluminum alloy is planned to be used for reverse circulation drilling in snow and ice. During drilling with reverse circulation, the compressed air or drilling liquid is injected through the inner and outer tube of the double-wall drill rod and returned to surface through the central passage of the inner tube. Air system is utilized to generate dry and cold compressed air. During snow and firn drilling, compressed air with reverse circulation was used, which can effectively prevent the leakage of compressed air into the surrounding snow. The process continued until the ice pressure at the depth posed a risk of downhole drill sticking, necessitating operational adjustments. Then, drilling liquid with reverse circulation was used to drill the ice below. In this way, the erosion of drilling liquid to the borehole wall can be avoided. Further, it may help in preventing possible hydraulic fracturing of ice borehole wall. Three types of drill bits made by steel, tungsten carbide and polycrystalline diamond compact (PDC) are prepared for ice drilling. Wireline coring drill with impregnated diamond drill bit is planned to be used by MPDS to obtain the subglacial bedrock.



Figure 1: Schematic diagram of the MPDS for subglacial bedrock sampling in Antarctica

This paper presents the design and test of the air system and the DFCS. Both systems were newly designed based on previous experiences. We believe that our experience and lessons will aid future drill design and promote the development of ice and subglacial bedrock drilling technology. In addition, the two systems are also helpful in preventing permafrost thawing during well drilling in permafrost regions and reduce the subsequent environmental threat (Eppelbaum and Kutasov, 2019; Langer et al., 2023).

## 2 Design of the air system and the DFCS

## 2.1 Design of the Air-system





## 2.1.1 Requirement of the air system

Theoretically, the required flow rate and pressure of compressed air is related to borehole depth and diameter, rate of penetration, size of drill pipes, size of ice chips and ice sheet temperature (Cao et al., 2018). According to Cao et al. (2019), a 5.25 Nm<sup>3</sup>·min<sup>-1</sup> flow rate and 1.06 MPa pressure were required to pneumatically transport an ice core with a 90 mm diameter and a 500 mm length from a 500 m depth to the surface. In general, higher flow rates and pressures of compressed air are better for drilling, but these require heavy compressors, which pose logistical challenges in remote Antarctic regions. The final compressed-air flow rate and pressure was determined to be no less than 10 Nm<sup>3</sup>·min<sup>-1</sup> and 1.5 MPa.

In theory, the compressed air should be, at least, cooled to below 0 °C to prevent ice from melting. We expected that a temperature of -5 °C would be better for cooling drill pipes and drill bit. The dew point of compressed air was expected to be lower than -40 °C. At the dew point of -40 °C, the water content in the compressed air drops to 0.176 g·m-3, which is considered to be very dry. The targeted drilling area of the MPDS has a distance less than 100 km away from Antarctic coast. In the targeted drilling area, the average atmosphere temperature is usually less than 30 °C (Wang and Hou, 2011). Consequently, the air system is also required to work at temperature of -30 °C and must be integrated into a standard 20 ft container for easy transportation on ice. In addition, the air system must have a modular design, and each module must weigh less than four tons for safe helicopter transportation from the icebreaker *Xuelong* to the ice surface. All the requirements of the air system are summarized in Table 1.

Table 1 Required design parameters of the air system

| Parameter                                    | Value                                       |  |
|----------------------------------------------|---------------------------------------------|--|
| Flow rate of compressed air                  | ≥10 Nm³·min⁻¹                               |  |
| Pressure of compressed air                   | ≥1.5 MPa                                    |  |
| Temperature of compressed air                | ≤-5 °C                                      |  |
| Dew point of compressed air                  | ≤-40 °C                                     |  |
| Low temperature resistance of the air system | ≤-30 °C                                     |  |
| Volume of the air system                     | Should be integrated into a 20 ft container |  |
| Waisht of the air avators                    | Should be separable into several unit       |  |
| Weight of the air system                     | weighing less than four tons                |  |

## 2.1.1 Principle of the air system

As shown in Fig. 2, the air system mainly contains a compressor, a receiver, a freezing dryer, a desiccant dyer, a cooler, a cyclone dust collector, three air filters and several sensors (Fig. 2). The compressor is used for generating compressed air

from atmosphere. When air at atmosphereic pressure is compressed to high pressures, its temperature can rapidly increase. To lubricate the compressor and to protect it from overheating, cold oil is usually pumped into the compressor and air fans are also used for cooling. When the air temperature drops, the water vapor in the air condenses, and a built-in gas-liquid separator is used to separate water or oil drops from the compressed air.

Figure 2: Principle of the air system





Then, the compressed air is flowed into a receiver to reduce the pressure pulses caused by the compressor. Here, the compressed air cools slightly owing to heat transfer with ambient atmosphere. During operation, the condensed water collects at the bottom of the receiver and is emptied periodically.

Subsequently, the compressed air is dried by the freezing dryer and desiccant dryers, respectively. The freezing dryer uses a heat exchanger to cool the compressed air by refrigerant. During the cooling process, the water vapor continues to condense and is separated from the air. Generally, the dew-point of compressed air can be lowered to 2–10 °C. To further dry the compressed air, a desiccant dryer with two dry towers is used. Different types of desiccants can be used to dry the air, such as silica gel, activated alumina, and molecular sieves. To lower the dew-point of the compressed air below -40 °C, molecular sieves are preferred. When a desiccant dryer is used, one tower works to dry the air and the other is heated to dry the desiccant. In modern design, the two towers can be automatically switched on periodically. The air filter is mainly used to remove impurities in the air, such as oil droplets, water droplets and micro solid particles. To protect the dryers, air filters 1 and 2 were inserted to remove the impurities in the compressed air. Overall, the air system employs a two-stage dehumidification process, offering greater reliability than conventional air-drying methods used in polar regions.

After passing through the desiccant dryer, the compressed air is sufficiently dry and could be cooled by air cooler. The air cooler uses heat exchanger to cool the air by refrigerant. Compared with air-cooling methods used in the past, using

refrigerant to cool compressed air ensures a consistently sub-zero temperature (

Figure 3: Main components of the air system: (a) air compressor; (b) air receiver; (c) freezing dryer; (d) desiccant dryer; (e) air cooler, 1-flowmeter, 2-dew-point hygrometer, 3-manometer, 4-temperature sensor; and (f) cyclone dust collector.

180

185

200

The freezing dryer utilize the R404A refrigerant to cool the compressed air and the refrigerant is air-cooled in condenser. The condensate could be discharged periodically by an electronic timed drain valve. The freezing dryer has built-in temperature sensors to monitor the air temperature at its inlet and outlet.

The desiccant dryer uses an alumina molecular sieve to absorb any leftover water vapor. The absorption tower is externally heated with power of 4.5 kW to heat the compressed air to 180–220°C. And then, the heated compressed air is used to dry the moist desiccant. Generally, it takes about 15–30 minutes to dry the moist desiccant in one absorption tower. The desiccant in the tower should be changed from time to time.

The air is cooled in two stages using two independently functioning coolers: first, to a temperature of -10–0°C, and then to below -10 °C. The same as freezing dryer, the air cooler also uses R404A to cool the compressed air by using a plate heat exchanger. The dew-point of the air fed into the air cooler has to be lower than -10°C. Otherwise, the water vapor would have condensed in the conduit and refrozen, leading to ice plugs. The dew-point at the inlet of the air cooler is always monitored by the built-in dew-point hygrometer. The temperature at the outlet of the first cooler is monitored in real time by a built-in temperature sensor.

Three air filters with different filtration ratings are used in the air system. As the compressed air flows from compressor to air cooler, the air filter precision increases. The air-filter drain valves are used to remove the filtered impurities. The sensors installed at the outlet of the air cooler can work in cold environment and can send data to a centralized control cabinet for data displaying. The cyclone dust collector is made by stainless steel and weighs less than 20 kg.

Table 2 Key parameters of the main components used in the air system

| Components     | Туре      | Key parameters                                     | Manufacturer  |
|----------------|-----------|----------------------------------------------------|---------------|
|                | 2LG-12/15 | Flow rate: 12 Nm <sup>3</sup> ·min <sup>-1</sup>   | Shanxi Yishan |
| Air            |           | Max. pressure: 1.5 MPa                             |               |
| compressor     |           | Power: 90 kW                                       | Compressor    |
|                |           | Weight: 1550 kg                                    | Co., Ltd.     |
| Air receiver C | C-1.0/1.6 | Design pressure: 1.68 MPa                          | Puyang Anuo   |
|                |           | Max. pressure: 2.1 MPa                             | Energy        |
|                |           | Air storage capacity: 1 m <sup>3</sup>             | Technology    |
|                |           | Weight: 371 kg                                     | Co., Ltd.     |
| Freezing       | YS-13T    | Flow rate: 13.5 Nm <sup>3</sup> ·min <sup>-1</sup> | Zhejiang      |
| dryer          |           | Working pressure: ≤1.6 MPa                         | Dongzhou      |

|              |                       | Required air temperature at inlet: ≤80 °C                    | Boiler Vessels |
|--------------|-----------------------|--------------------------------------------------------------|----------------|
|              |                       | Dew point of air after drying: 2-10 °C                       | Co., Ltd.      |
|              |                       | Power: 2.3 kW                                                |                |
|              |                       | Flow rate: 13.5 Nm <sup>3</sup> ·min <sup>-1</sup>           |                |
|              |                       | Working pressure: ≤1.6 MPa                                   |                |
| Desiccant    | YS-13R                | Adsorbent: Alumina molecular sieve                           |                |
| dryer        |                       | Required air temperature at inlet: ≤40 °C                    |                |
|              |                       | Dew point of air after drying: -20–40 °C                     |                |
|              |                       | Heating power: 4.5 kW                                        |                |
|              |                       | Flow rate: 13.5 Nm <sup>3</sup> ·min <sup>-1</sup>           |                |
|              |                       | Working pressure: ≤1.6 MPa                                   |                |
| Air filter 1 | YS-013T               | Working temperature: ≤66 °C                                  |                |
|              |                       | Filtration rating: 1 μm                                      |                |
|              |                       | Oil remainder: 3 ppm                                         |                |
|              |                       | Flow rate: 13.5 Nm <sup>3</sup> ·min <sup>-1</sup>           |                |
|              |                       | Working pressure: ≤1.6 MPa                                   |                |
| Air filter 2 | YS-013H               | Working temperature: ≤66 °C                                  |                |
|              |                       | Filtration rating: 0. 1 μm                                   |                |
|              |                       | Oil remainder: 2 ppm                                         |                |
|              |                       | Flow rate: 13.5 Nm <sup>3</sup> ·min <sup>-1</sup>           |                |
|              |                       | Working pressure: ≤1.6 MPa                                   |                |
| Air filter 3 | YS-013A               | Working temperature: ≤66 °C                                  |                |
|              |                       | Filtration rating: 0.01 μm                                   |                |
|              |                       | Oil remainder: 1 ppm                                         |                |
|              |                       | Flow rate: ≤11 Nm <sup>3</sup> ·min <sup>-1</sup>            | Xi'an          |
|              |                       | Working pressure: ≤1.5 MPa                                   | Hengmao        |
| Air cooler   | FDK-10/10-C           | Air temperature after cooling: -10–0 °C                      | Power          |
|              |                       | Power: 20 kW                                                 | Technology     |
|              |                       | Weight: 795 kg                                               | Co., Ltd.      |
|              | ONEC                  | Measurement range: 300–3600 Nm <sup>3</sup> ·h <sup>-1</sup> | Shaanxi OPEC   |
| Flowmeter    | OPEC-<br>LUGB2G04CYNE | Max. working pressure: 4 MPa                                 | Instrument     |
|              |                       | Precision: ±1.5 %;                                           | Group Co.,     |
| Manometer    | OPEC-                 | Measurement range: 0-2.5 MPa                                 | Ltd.           |
|              |                       |                                                              |                |

|              | 2088SC8M1NM1E | Precision: ±0.5 %;                               |            |
|--------------|---------------|--------------------------------------------------|------------|
| Temperature  | OPEC-         | Measurement range: -40-150 °C                    |            |
| sensor       | SBWZP1KS51YE  | Precision: ±0.5 %;                               |            |
|              |               | Measurement range: -60-60 °C                     |            |
| Dew-point    | ODEC DM70EC/N | Precision: ±0.1 °C                               |            |
| hygrometer   | OPEC-DM70ES/N | Max. working pressure: 4 MPa                     |            |
|              |               | Max. working temperature: 70 °C                  |            |
|              |               | Flow rate: 10 Nm <sup>3</sup> ·min <sup>-1</sup> |            |
| Cyclone dust | 1             | Max. pressure: 1.5 MPa                           | Customized |
| collector    | /             | Diameter: 300 mm                                 | Customized |
|              |               | Height: 1.83 m                                   |            |

# 2.1.4 Integration of the air system into the container



All the components mentioned in Sect. 2.3.1 were integrated into a customized 20 ft container with left and right parts, connected by bolts (Figs. 4). In this paper, the left part refers to the half container close to the triangular bracket of the sled, while the right part refers to another half container. The compressor, receiver, and centralized control cabinet were installed in the left part. To prevent container overheating by the waste heat generated by compressor, a thermally insulated cover was added on the top of air fan which guided the hot air to the outside of the container through a window shutter. In addition, there were cooling vents on the front door of left container. The centralized control cabinet controlled the component start and stop and displayed the data collected by the sensors. The operator entrance was a small door near centralized control cabinet. All the components could also be started or stopped by their own control boards.

Figure 4: Structure of the air system

The freeing dryer, desiccant dryer, air cooler, filters and sensors were placed in the right part. The freeing dryer and desiccant dryer were installed to one side and the air cooler to the other side. A small sidewalk between the two sides facilitated the passage of a single person. The air cooler also had a fan cover for heat dissipation through a window shutter on the right container. All the window shutters and the cooling vents had thermally insulated covers to keep the container to prevent snow from entering when the air system was not in use.

All the components were fixed by bolts on the container floor. The total weight of the air system was approximately 7.8 tons. During transportation to Antarctica, the air system could be disassembled into left and right parts. The left part weighed 4.2 tons, while the right part weighed ~3.6 tons. The two parts were reassembled on the field using bolts; the gap between them could be sealed using a sealant to prevent rain and snow from entering. On ice surface, the air system could be easily moved by a sledge. As the compressor and freezing dryer could only be started up above 0 °C, 2 kW heaters were installed in both the left and right parts. Before starting the whole air system, the heaters were used to warm the container. Once the system starting to work, the heaters would be shut off. Meanwhile, the thermally insulated covers of the cooling vents and window shutter were removed for heat dissipation.

## 2.2 Design of the DFCS






## 2.2.1 Requirement of the DFCS

DFCS requirements are shown in Table 3. The flow rate and pressure required for ice and subglacial bedrock drilling below 1000 m was roughly estimated according to the method of Alemany et al. (2021). The final determined flow rate for the DFCS is a little lower than the maximum flow rate of 134 L·min<sup>-1</sup> used by RAID (Goodge and Severinghaus, 2016). To prevent possible hydraulic ice fracture caused by drilling-liquid overpressure, the DFCS maximum pressure was limited to less than 2 MPa. The drilling fluid must be colder than 0 °C in Antarctica before being pumped into a borehole to prevent melting of the borehole ice wall. In our case, the temperature of drilling liquid is expected to be lower than -2°C. The DFCS has the same requirements as the air system in low-temperature resistance, volume and weight.

Table 3 Required design parameters of the DFCS

| Parameter                              | Value                                     |  |
|----------------------------------------|-------------------------------------------|--|
| Flow rate of drilling fluid            | ≥100 L·min-1                              |  |
| Maximum pressure of drilling fluid     | ≤2 MPa                                    |  |
| Temperature of drilling fluid          | ≤-2 °C                                    |  |
| Low temperature resistance of the DFCS | ≤-30 °C                                   |  |
| Volume of the DFCS                     | Should be integrated in a 20 ft container |  |
| Weight of the DFCS                     | Should be separable into several units    |  |

## 2.2.2 Principle of the DFCS




As shown in Fig. 5, the DFCS uses a refrigerating machine to cool the drilling fluid at the surface. The refrigerating machine could cool the refrigerant to a low temperature and then pump it into a heat exchanger by a built-in pump. In the heat exchanger, the drilling fluid is cooled to desired temperature by the cold refrigerant. Generally, a circulation pump is used to pump the drilling liquid from circulation tank to the heat exchanger. Different types of heat exchangers were considered in the design, including the plate heat exchanger, finned-tube heat exchanger, shell and tube heat exchangers. Compared with finned-tube and shell and tube heat exchangers, the plate heat exchanger is more compact in size and lighter in weight. There are two types of plate heat exchangers, which are the brazed plate heat exchanger and gasket-type plate heat exchanger. The brazed plate heat exchanger is difficult to clean when solid particles accumulate inside. A gasket-type plate heat exchanger was selected because it uses a bolt-connected plate for easy disassembly during cleaning. In this way, the drilling fluid can be actively cooled to a low temperature, preventing the melting of the ice-borehole wall—a phenomenon observed in other subglacial bedrock drilling projects. The drilling liquid don't need to be actively cooled if the ambient temperature is low enough to cool the drilling liquid to a temperature below -2 °C.

Figure 5: Principle of the DFCS

To make it lighter for easy transportation, the tank for drilling-liquid storage is partitioned into a circulation tank and a stirring tank. The drilling liquid is prepared in the stirring tank, and a stirrer is installed inside to evenly mix the different additives and base fluid. The circulation tank and the stirring tank are hydraulically connected by a pipeline. To monitor the liquid level in the two tanks, liquid-level gauges are installed above the tanks. The prepared drilling fluid is then injected

from the stirring tank to the borehole by a drilling-fluid pump. At the outlet of the drilling-fluid pump, a flowmeter, manometer, and temperature sensor are installed to monitor the key parameters of the drilling liquid.

Vibration sieves and vertical centrifuges are used for separating ice or rock cuttings from the drilling liquid. When the drilling liquid returns to surface, it is stored in a settling tank and then pumped through a vibration sieve for separating the ice or rock cuttings. The vibration sieve is installed on the circulation tank, in which the drilling fluid could be stored after separation. The separated ice or rock cuttings are still wet with drilling liquid, so a vertical centrifuge is used to dry them. The residual drilling fluid in the ice or rock cuttings is recycled and only the dried ice and rock cuttings are discarded. Comparing with the heated melter tank used by RAID and ASIG, the vertical centrifuge is more efficient and energy-saving (Kuhl et al., 2021; Goodge et al., 2021).

## 265 **2.2.3** Components of the DFCS



All the components of the DFCS were selected based on the requirements and were commercially available, except for the stirring, circulation, and settling tanks (Fig. 6). The key parameters of the components are shown in Table 4. The refrigerating machine cools the secondary refrigerant using R404A. The secondary refrigerant is then pumped to the heat exchanger to cool the drilling liquid. In the refrigerating machine, glycol aqueous solution with 60 % volume fraction and freezing point of -50 °C acts as secondary refrigerant. The refrigerating machine could generate a lot of heat during operation, so two large air fans are installed at the back for heat dissipation.

Figure 6: Main components of the DFCS: (a) refrigerating machine; (b) heat exchanger; (c) drilling-fluid pump; (d) vibration sieve; (e) vertical centrifuge; (f) circulating pump; and (g) stirrer.





The plate in the gasket-type plate heat exchanger is made by stainless steel 316, which has a thickness of 0.5 mm. The plate gap allows solid particles smaller than 1 mm to pass through.

A three-cylinder plunger pump, driven by an AC motor using a cold-resistant rubber belt, was used as the drilling-fluid pump. To reduce the pulse pressure generated by the three cylinders, an accumulator was installed at the pump outlet. A pressure-relief valve was installed to regulate pressure and flow rate. All seals and the pistons in the pump are made of low-temperature resistant rubber, which can work at -30 °C. Vertical centrifugal pumps were chosen as circulation pump. The centrifugal pumps can be used to pump organic substances such as aviation kerosene and silicone oil at -30 °C.

The chosen vibration sieve can generate linear motion on the cuttings and have a single layer of mesh. The mesh has an area of  $\sim$ 2 m<sup>2</sup> and can be replaced based on the cuttings size. The vibration sieve was installed above the circulation tank with four springs. The separated cuttings can slide into the vertical centrifuge along a guide groove, fixed at an angle on the circulation tank. The controls for starting and stopping the vibration sieve, drilling-fluid pump, and vertical centrifugal pump are accessible on the centralized control cabinet.

The vertical centrifuge is powered by a variable-frequency motor, with a maximum rotation speed of 1500 r/min. The drilling fluid remaining in the cuttings is separated by a replaceable filter screen installed in the circular drum and then pumped back to the circulation tank. The dry cuttings can be only manually removed from the drum by the operator when connecting additional drill pipe to the drill string to extend its length. Four large springs were installed at the foot of the vertical centrifuge to reduce its vibration during high-speed rotation. In addition, the motors and the variable-frequency drive are covered by steel to prevent the drilling liquid from entering. The vertical centrifuge could be controlled using its own controller or by the centralized control cabinet.

The stirrer is vertically installed above the stirring tank and has only one layer of plate-type blades. The size of both the stirring and circulation tanks is  $1.3 \times 1.3 \times 1.2$  m. Therefore, the largest drilling-liquid volume that could be prepared at the surface is about 4 m<sup>3</sup>. Two settling tanks sized  $2 \times 1 \times 0.8$  m are used. The tanks are made of 3-mm steel plates and reinforced with square steel tubes.

Radar level gauges is installed on both tanks for real-time liquid-level measurement. The data measured by the radar level gauge, flowmeter, manometer, and temperature sensor are displayed on the centralized control cabinet and sent to the control panel of the drill rig.

Table 4 Key parameters of the main components used in the DFCS

| Components       | Type               | Key parameters                                         | Manufacturer    |
|------------------|--------------------|--------------------------------------------------------|-----------------|
|                  |                    | Nominal refrigerating power: 20.1 kW                   | Shenzhen Polyde |
| Refrigerating    | DI D 25 A I        | Total power: 23 kW                                     | Refrigeration   |
| BLD-25AL machine | Refrigerant: R404A | Technology Co.,                                        |                 |
|                  |                    | Secondary refrigerant: Glycol aqueous solution by 60 % | Ltd.            |

#### volume

|                     |           | volume                                                                            |                        |
|---------------------|-----------|-----------------------------------------------------------------------------------|------------------------|
|                     |           | Weight: 750 kg                                                                    |                        |
|                     |           | Flow rate of pump for secondary refrigerant: $15 \text{ m}^3 \cdot \text{h}^{-1}$ |                        |
|                     |           | Head of pump for secondary refrigerant: 30 m                                      |                        |
|                     |           | Tank volume for secondary refrigerant: 200 L                                      |                        |
|                     |           | Number of plates: 20                                                              | Alfa Laval             |
|                     |           | Maximum heat-transfer power: 46.9 kW                                              |                        |
| Heat exchanger      | T6-PFG    | Heat-exchange area: 7.2 m <sup>2</sup>                                            | (Shanghai)             |
|                     |           | Design pressure: 1 MPa                                                            | Technologies Co.,      |
|                     |           | Weight: 163 kg                                                                    | Ltd.                   |
|                     |           | Power: 7.5 kW                                                                     | W/: T: II:-1           |
| D.'II' (I' I        | 2CD40 2/6 | Max. outlet pressure: 2 MPa                                                       | Wuxi Terui High        |
| Drilling-fluid pump | 3SP40-2/6 | Displacement: 6–7 m <sup>3</sup> ·h <sup>-1</sup>                                 | Pressure Pump          |
|                     |           | Weight: 370 kg                                                                    | Valve Co., Ltd.        |
|                     |           | Processing capacity of drilling fluid: 8 m <sup>3</sup> ·h <sup>-1</sup>          | Xi 'an Xingqing        |
| Vibration sieve     | ZZSD-01   | Power: 0.25 kW×2                                                                  | Petroleum              |
| vioration sieve     |           | Number of the mesh: 40–150                                                        | Equipment Co.,         |
|                     |           | Weight: 150 kg;                                                                   | Ltd.                   |
|                     |           | Power: 4 kW                                                                       | 7h an aile ann a       |
|                     | PSL600    | Separation factor: 755                                                            | Zhangjiagang<br>Guohua |
| Vertical centrifuge |           | Max. rotation speed: 1500 r·min <sup>-1</sup>                                     |                        |
|                     |           | Working volume: 38 L                                                              | Machinery Co.,         |
|                     |           | Weight: 700 kg;                                                                   | Ltd.                   |
|                     |           | Power: 1.1 kW                                                                     | Ch h - : C h :         |
| Circulating pump    | CDLF4-5   | Flow rate: 6 m <sup>3</sup> ·h <sup>-1</sup>                                      | Shanghai Sunshine      |
|                     |           | Head: 31m                                                                         | Pump Co., Ltd.         |
|                     |           |                                                                                   | Cangzhou               |
|                     |           | Rotation speed: 84 r·min <sup>-1</sup>                                            | Huacang Drilling       |
| Stirrer             | WHNJ-1.5  | Power: 1.5 kw                                                                     | & Extracting Oil       |
|                     |           | Weight: 50 kg                                                                     | Equipment Co.,         |
|                     |           |                                                                                   | Ltd.                   |
| Liquid lavel cours  | 80C       | Measurement range: 0.08-10 m                                                      | Hangzhou Supmea        |
| Liquid-level gauge  | 80G       | Precision: ±3 cm                                                                  | Automation Co.,        |

|                    |           |                                                          | Ltd.            |
|--------------------|-----------|----------------------------------------------------------|-----------------|
| Flowmeter          | LZ-FK-    | Measurement range: 0.8–8 m <sup>3</sup> ·h <sup>-1</sup> |                 |
|                    | DN40      | Precision: ±1.5 %                                        | Hangzhou        |
| Manometer          | FK-P400-E | Measurement range: 0–5 MPa                               | FULLKON         |
|                    |           | Precision: ±0.5 %                                        | Instrument Co., |
| Temperature sensor | FK-WZPK-  | Measurement range: -200-350 °C                           | Ltd.            |
|                    |           | Precision: ±1.5 %                                        |                 |

## 2.2.4 Integration of the DFCS into the container



All the components of the DFCS were integrated into a standard 20 ft container, except for the two settling tanks (Fig.7). The standard container was partitioned into right and left containers, which could be connected by bolts. The refrigerating machine, the heat exchanger, circulation pump 1, and the stirring tank with a stirrer on top were installed in left container. The left container had front and side doors. The front door had two cooling vents for refrigerating-machine heat dissipation during operation. The side door was close to the heat exchanger. The side-door entrance had sufficient space for operating the refrigerating machine and preparing the drilling liquid in the stirring tank.

Figure 7: Structure of the DFCS

The rest of the components were installed in the right container. The circulation tank, vibration sieve, and vertical centrifuge were fixed on one side. The drilling-fluid pump, circulation pump 2, and the sensors were installed on another side. The vibration sieve was installed above the circulation tank, and a guide groove was used to transport the separated cuttings from the vibration sieve to the vertical centrifuge. The right container had a small door and a back door. The small door was close to the centralized control cabinet for easy operation.

All the components were fixed to the floor of the container by bolts. Each half container with all its components inside weighed about 3.7 tons. A 2 kW heater was installed in each half container. A centralized control cabinet was used to control all the components and display the measured data from sensors. To save space, the centralized control cabinet was divided into two parts, the first was fixed on the stirring tank, and the second was fixed on the circulation tank. The settling tank was lower than borehole mouth and below the snow surface in the field. The returned drilling fluid can flow from borehole mouth into the settling tank owing to gravity.

#### 3 Testing of the air system and the DFCS

#### 3.1 Testing of the air system







#### 3.1.1 Performance of the air system

The air system was first tested in Zhangjiakou, Hebei Province, China after manufacturing. The testing site is about 200 km away from Beijing. In domestic tests, the air system worked for two hours and generated 10.9 Nm³·min⁻¹ compressed air at 1.5 MPa pressure. At an ambient temperature of 27–28 °C, the air temperature rose to 95–97 °C post-compression. The compressed air was cooled to -4.5 °C with first cooler and to -10.8 °C with both the first and the second coolers. After drying using the freezing dryer and desiccant dryer, the dew-point of the compressed air was as low as -48 to -52 °C. After testing, the air system was used to drill the underground soil and rock using the drill rig. However, because of groundwater presence, only the compressor and receiver were used. The freezing dryer, desiccant dryer, and cooler did not test. The drill-rig testing using the compressor, receiver, and cyclone dust collector lasted for approximately two months, during which they performed stably. It is worthy to be mentioned that the testing environment in China is quite different with Antarctica and the domestic testing can only check the basic function of the air system.

The MPDS was tested in the summer season of 2024/2025, at the margin of Princess Elizabeth Land, East Antarctica (coordinates: 69°35'10.12" S, 76°23'03.87" E). The testing site is only about 50 m away from the drilling site of the Russian-Chinese drilling project in 2023/2024 season. The ice sheet thickness at the drilling site is about 545 m according to the drilling results (Leitchenkov et al., 2024). In the field test, a 110 m borehole was drilled, in which upper the 42 m were drilled using the air system. The upper 42 m of the borehole was drilled by a 178 mm drill bit made by steel. Total 25 days were spent to deepen the hole from surface to 42 m because of many unexpected problems in the drill bit and air the system. The air system began work on January 10<sup>th</sup>, 2025. The testing results of the air-system from January 12<sup>th</sup> to 15<sup>th</sup>, 2025 are shown in Fig.8. The flow rate of compressed air was 9.3–12.4 Nm³·min⁻¹ with an average value of 10.5 Nm³·min⁻¹ (Fig.8a). The average pressure of compressor was kept at 0.7–1 MPa, owing to the pressure loss at the conduit of the surface equipment (freezing dryer, desiccant dryer, air cooler, air hose, swivel, valves etc.) and downhole drill tools (Fig.8b). The air temperature increased to approximately 76–89 °C with an average value of 81 °C after compressed by the compressor (Fig.8c). After cooling by the built-in air fans in the compressor and the receiver, the compressed air entered the freezing

dryer at a temperature of 24-44 °C (Fig.8d). However, the freezing dryer did not work well, and the compressed air was cooled only to an average temperature of ~35 °C, and the compressed air was not dried to the required dew-point of 2–10 °C (Fig.8e). In the air cooler, the compressed air was cooled again. In field testing, only one stage of cooling was used. When only the first cooler was used, the air temperature at the its outlet decreased to-17.4-7.9 °C (Fig.8f). However, the temperature of the compressed air increased to -12.4--4.9°C after flowing through the second cooler (Fig.8g). The reason is that the second cooler did not work and the compressed air was warmed a little by ambient atmosphere in heat exchanger of the cooler. The first cooler was not used after January 14th, When only the second cooler was used for air cooling, the temperature sensor installed at the outlet of the first cooler monitored a temperature variation from 17.6 to 41.7 °C (Fig.8h). After cooled by the second cooler, air temperature can decrease to -15.9-5.1 °C (Fig.8i). Regardless of whether the first or second cooler is utilized, the temperature of compresses air was lower than to -4.9 °C before injected into the ice borehole. In general, the air cooler can cool the compressed air effectively and its cooling capacity is comparable to that of RAM-2 drill (Gibson et. al. 2021). As shown in Fig. 8i, the dew-point of the compressed air varied from -11.3 to 19.9 °C after dried by the freezing dryer and the desiccant dryer. However, the dew-point of the compressed air was higher than 0 °C in most situation, which was significantly below the required dew point of -40°C. The testing results showed the poor performance of the freezing dryer and the desiccant dryer. After cooled in the cooler, the dew point of the compressed air decreased to -21.2-4.6 °C (Fig. 8k). This implies that some water vapor has been frozen into ice in the cooler. The frozen ice can accumulate in the pipeline of the cooler and then lead to ice plug. That is the reason why the air cooler was broken after January 15th. The detailed data regarding to the testing results can be found in the corresponding supplements Table S1.





## Figure 8: Test results of the air system in Antarctica




As mentioned above, the freezing dryer and desiccant dryer did not work well, and the compressed air could not be dried to a dew-point of -40 °C before flowing through air cooler. In this case, the condensate froze into an ice plug in the air-cooler conduit leading to a rapid rise in the outlet pressure of the air cooler. In common case, after 15–20 minutes, the pressure increased from 0.7 MPa to 1.3 MPa, which was close to the limited maximum pressure of air cooler. In the field, the ice plug was melted by hot compressed air after shutting off the air cooler. In this situation, the drilling work had to be stopped until the pressure dropped to normal values. This process took approximately 5–8 minutes. In addition, another 3–5 minutes were required to restart the cooler. Frequent ice plug formation in the air-cooler conduit significantly hinders continuous ice drilling operations, drastically reducing drilling efficiency. In some cases, drilling must be interrupted up to three times within a single run while waiting for the air system to be restored to operational readiness.

In the field, the first and second coolers were used alternately, and the operators spent a lot of time on monitoring the pressure, starting and shutting off the cooler etc. However, the heat exchanger of the first air cooler was accidentally broken by frozen ice on January 16<sup>th</sup>, and the air cooler stopped working. Later, all the components were carefully checked in the field. It was found that a lot of the condensate was not discharged from the air receiver through the drain valve (Fig. 9a). We suspect it is one of the reasons why the freezing dryer and the desiccant dryer could not dry the compressed air to desired dew-point. Additionally, lot of the condensate was also found in the freezing dryer and the air filters, which share the drainage channel. It is suspected that the drainage channel was blocked somewhere or the drain valve on the freezing dryer and the air filters stopped to work. Manufacturing defects of the freezing dryer and the desiccant dryer were also suspected. Due to limited working time in Antarctic field, the air system was not carefully checked and the specific reasons for the failure of the freezing dryer and the desiccant dryer remains unclear.

To continue drilling with air, a 140 m rubber hose was buried approximately 10 cm deep in the snow to cool the compressed air (Fig. 9b). This method proved to be partly useful. In Antarctica, compressed air at a temperature of 15–21 °C could be cooled to -1 to -3 °C with an ambient temperature of about -6 °C. However, this method could not decrease the humidity of compressed air, and the condensate refroze on the outer surface of the drill pipe and also forming an ice plug inside (Figs. 9c and 9d). Overall, long-term drilling with compressed air in Antarctic without freezing dryer and the desiccant dryer is unrealistic.

Figure 9: Test phenomena: (a) cyclone dust collector worked well in domestic test; (b) condensed water could not be discharged from the receiver; (c) burying the rubber hose into snow to cool compressed air; (d) condensed water refrozen on the outer surface drill pipe; (e) ice plug formed in drill pipe.

## 400 3.1.2 Suggestions for improvement of the air system

The modular design of the air system had some good features. In the field, it could be easily assembled and disassembled. In addition, all components of the air system could meet the design requirements in domestic tests, particularly, the compressor, air cooler, and cyclone dust collector. However, the poor performance of the freezing dryer and the desiccant dryer in the field led to air-cooler breakage and refrozen water on the drill pipes.

In future, the discharge pathway of condensate in receiver must be redesigned. Besides, freezing dryer and desiccant dryer of good quality must be used. The broken heat exchanger in the cooler must be replaced. A passive heat exchanger is suggested for cooling compressed air in Antarctica where the ambient temperature is lower than -20 °C, because passive heat exchangers are simpler, lighter and more reliable than the air cooler (Bently, et al., 2009; Talalay and Pyne, 2017).

## 410 3.2 Testing of the DFCS

## 3.2.1 Performance of the DFCS

The DFCS was tested both in China and in Antarctica. The domestic test used water, and the maximum flow rate of drilling-fluid pump was as high as 110 L·min<sup>-1</sup>. The vibration sieve and the vertical centrifuge were tested with mud, containing both

soil and sand. In general, the vibration sieve and the vertical centrifuge worked well, and the solid particles could be separated from water. To test the capability of the refrigerating machine and the heat exchanger, clean water was circulated at flow rate of 100 L·min<sup>-1</sup>. In the test, 2 m<sup>3</sup> water could be cooled from 24.8 to 7 °C in 50 minutes.

The DFCS was used to drill an ice borehole in Antarctica from 31–110 m in the summer season of 2024/2025. The diameter of the borehole from 31m to 110 m was 102 mm and a production drilling rate was about 8–12m/day. In the field test, the flow rate of the drilling-fluid pump was controlled to 62–96 L·min<sup>-1</sup> while the pressure drop was maintained at 0.15–0.29 MPa (Fig. 10a and 10b). The drilling fluid was kept at approximately -15.5–4.5 °C at surface because of low ambient atmosphere temperature, so the refrigerating machine and the heat exchanger were never used in the field (Fig. 10c). The detailed data regarding to the Figure 10 can be found in the supplements Table S1.




Figure 10: Test results of the DFCS in Antarctica

The 100 meshes and 120 meshes screens were tested for separating ice cuttings (Fig. 11). In general, the 100-mesh screen worked much better than the 120-mesh screen. When the 100 meshes screen was used, the ice chips aggregated into small balls on the screen, and most of the drilling liquid leaked into the circulation tank below. However, some very fine ice chips could not be filtered from the drilling liquid with 100 meshes screen. The vertical centrifuge was operated at its highest rotation speed in the field. When the ice-chip balls fell into the vertical centrifuge, almost all the ice chips could be removed from the drilling liquid. The 120 meshes screen was much better in removing fine ice chips. However, the ice chips and the clay additives in the drilling liquid clogged the small holes on the screen, and some of the drilling liquid did not reach the circulation tank; some ice chips remained in the drilling fluid even after filtering by the vertical centrifuge.

Figure 11: Performance of vibration sieve and vertical centrifuge: (a) small ball of ice chips formed on 100 meshes screen; (b) cleaned drilling fluid when using 100 meshes screen; (c) clogging of 120 meshes screen by ice chips and clay additives; (d) ice chips remaining in the drilling liquid when using 120 meshes screen.

In the Antarctic test, other problems were found with the DFCS. The drilling-fluid pump could not be controlled by frequency modulation. In the field, the flow rate of drilling liquid had to be regulated with a pressure-relief valve by operator. Besides, the circulation pump could not suck the drilling liquid from settling tank to vibration sieve because the liquid level in settling tank was lower than the circulation pump. The ice chips became very hard after being centrifuged. Removal of the hard ice chips from centrifuge drum was difficult and labor-intensive. In addition, there was no brake on the vertical centrifuge and much time was wasted in waiting for it to stop before cleaning it. The circulation tank had no stirrer, and the clay added in the drilling liquid settled down after a while. Also, the settled clay could get pumped into heat exchanger and block the flow channels, as there was no filter screen between circulation pump 1 and the circulation tank.

## 3.2.2 Suggestions for improvement of the DFCS



In general, the DFCS was reasonably designed and met the basic usage requirements. However, some problems remained. Here, the following suggestions are given for future improvement of the system. First, the flow rate of the drilling-fluid

pump should be adjustable by variable-frequency drive for easy operation. Otherwise, the drilling-fluid pump has to be manually operated and can easily lead to mismatch of flow rate with drilling requirement. Second, the pump with suction capacity should be used as circulation pump, such as self-priming pumps and submersible pumps. Third, a high-capacity vibration sieve with double layer of screens is suggested to improve its processing capacity of drilling fluid. Fourth, a type of vertical centrifuge with built-in brake should be used and the vertical centrifuge should have the ability to remove the hard ice chips by itself, instead of operator, to reduce the intensity of manual labour. Additionally, small stirrer with size less should be added on the circulation tank, and a filter screen made by steel wire are strongly suggested to add between the circulation pump and the heat exchanger to prevent the settled clays block the flow channel.

### 4 Experiences and lessons

In this study, an air system and a DFCS for subglacial bedrock sampling in Antarctica were designed and tested, leading to a gain in experience and lessons.

- (1) Standard 20 ft containers were successfully partitioned into two. The two half containers integrated with components almost weighed the same and could be easily transported and assembled in the field.
- (2) The testing results shows that the air system requires significant modifications and additional testing. In future designs, the compressed air must be dried to a dew point of less than -40 °C using better dryers before being cooled by the air cooler.
- Burying a long hose under snow appears to be an effective way of cooling compressed air in Antarctica. However, this method is only suggested for drilling shallow holes in a region with cold surface-air temperatures. For atmospheric temperatures lower than -20 °C, the feasibility of cooling compressed air by a passive heat exchanger with an air fan is worthy to be studied.
- (3) The refrigerating machine and heat exchanger were proved to be useless in field test because of low temperature at this drilling site; however, they could be helpful in warmer Antarctic regions. The possibility they would be helpful in cooling the drilling bits during subglacial rock drilling must be verified by future theoretical research or practical drilling engineering. As the variation of drilling-fluid temperatures during subglacial rock drilling is still unknown, a prediction model of the drilling-liquid temperature in reverse circulation is an urgent need.
- (4) A flow rate of 79.6 L·min<sup>-1</sup> and pressure of 0.18 MPa have been proven sufficient to drill a 110 m ice borehole with no hydraulic fracturing events. The vibration sieve and the vertical centrifuge required significant modifications to separate ice cuttings from the drilling liquid efficiently. In our opinion, a vertical centrifuge with a self-cleaning function would be better than the melter used by RAID and ASIG for further separation of drilling fluid from the ice chips, owing to efficiency and power savings.

Using the aforementioned experiences and lessons, improvements are planned to the DFCS and the air system. In the next summer season, the two systems are expected to drill a 1000 m borehole in Antarctica.

## Data availability

The testing data of the two system can be found in supplement material. Other data can be provided by the corresponding authors upon request.

## **Author contribution**

YG designed the project; LYa, YG designed the two system with the contribution from WZ and HR; LYa, YG, WJ, ZK, LB, LYan, ZZ, LX, SYu and WM carried out domestic test; LYa, ZK, LYan, ZZ, LX and SY carried out field test in Antarctica. LYa wrote the manuscript draft; YG, LB and ZK reviewed and edited the manuscript.

#### **Competing interests**

The authors declare that they have no conflict of interest.

#### 490 Acknowledgement

We thank all teachers in China University of Geosciences, Beijing for their helpful advice in designing of the air system and DFCS system. We also thank the members of Shaanxi Taihe Intelligent Drilling Co., Ltd involved in manufacturing of the two systems. We are grateful to CHINARE for logistical support of field operations in Antarctica, especially thanks to Zhongshan station and aircraft team for the continue support of snow vehicles.

#### 495 Financial support

This research has been supported by the National Key Research and Development Program of China [grant number 2021YFA0719104]; and the National Natural Science Foundation of China [grant number 42206255].

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
