# Peer review of "Experiences and Lessons Learned from Designing and Testing of an Air System and a Drilling Fluid Circulation System Adapted for Subglacial Bedrock Sampling in Antarctica"

_EGUsphere, 2025_

## Author Comment (AC1)

**Reply to the comments of referee #1**

The manuscript left me with mixed feelings. On the one hand, it contains interesting information about ice-drilling design and experiments. However, on the other hand, the information is not presented in a clear and logical sequence, and some important details are omitted. I believe that the article would benefit from major revisions before publication.

Thanks for your helpful comments. The manuscript has been carefully revised according to your suggestion.

**General comments:**

1. The title of the paper is too long and not very descriptive.

The title has been changed to "Experiences and Lessons Learned from Designing and Testing of an Air System and a Drilling Fluid Circulation System Adapted for Subglacial Bedrock Drilling in Antarctica". In the new title, the "experiences and lessons" is emphasized, and the two systems are emphasized to be used for subglacial bedrock drilling in Antarctica.

2. There is no information about the key technological elements of the drilling technology, such as the drill rig, drill pipes, drill bits, drilling fluid components, etc. Also, there is no information about the locations of the domestic and Antarctic test sites. Without this information, it is impossible to fully understand the process of drilling and problems.

The following sentences are added in the "Introduction" section to give more information about MPDS.

 "*All the subsystems are modularly designed and can be disassembled into several parts with each one less than 4 tons for easy transportation by helicopter from ice breaker to polar ice sheet. Once arrived at ice sheet, all the subsystems can be integrated in a 20'' container and can be transported on ice sheet by sledge. All the subsystems were integrated in standard 20 ft container and could be easily moved on the ice surface by sledges. The drill rig is fully driven by a hydraulic system and can work with different drilling processes, such as air/drilling liquid reverse circulation drilling and wireline coring drilling. The drill-rod module is used for drill rod storage and is adjacent to the drill rig for easy transferring of drill rod in field. Air system is utilized to generate dry and cold compressed air*"

"*Double-wall drill rod made by aluminum alloy is planned to be used for reverse circulation drilling in ice. Three types of drill bits made by steel, tungsten carbide and polycrystalline diamond compact (PDC) are prepared for ice drilling. Wireline coring drill with impregnated diamond drill bit is planned to be used by MPDS to obtain the subglacial bedrock*".

More details about the MPDS will be presented in our coming paper, which is now being prepared. The locations of the domestic and Antarctic test sites are added in revised manuscript. The added sentences are "*The air system was first tested in Zhangjiakou, Hebei Province, China after manufacturing. The testing site is about 200 km away from Beijing*", "*The testing site is only about 50 m away from the drilling site of the Russian-Chinese drilling project in 2023/2024 season. The ice sheet thickness at the drilling site is about 545 m according to the drilling results (Leitchenkov et al., 2024)*".

*References*

*Leitchenkov, G.L., Talalay, P.G., Zhang, N., Abdrakhmanov, I.A., Gong, D., Liu, Y., Li, Y., Sun, Y., Vorobyev, M., Li, B.: First targeted geological sampling beneath the East Antarctic ice sheet: joint Russian-Chinese drilling project. Exploration & Protection of Mineral Resources, Special issue, 75–78, 2024.*

3. The structure of the paper is not ideal. For instance, the "components" section includes "principles" as well. I would suggest reorganizing the structure of the paper in a more common way, with an introduction, methods (both air and drilling fluid), results (tests), and conclusions.

The structure of the manuscript was reorganized as the following way: introduction, design of the air system and the DFCS, testing of the air system and the DFCS, experiences and lessons.

In the revised manuscript, the related sentences in the "components" section have been moved to the "principles" section, such as "*The air filter is mainly used to remove impurities in the air, such as oil droplets, water droplets and micro solid particles.*", "*Different types of heat exchangers were considered in the design, including the plate heat exchanger, finned-tube heat exchanger, shell and tube heat exchangers. Compared with finned-tube and shell and tube heat exchangers, the plate heat exchanger is more compact in size and lighter in weight. There are two types of plate heat exchangers, which are the brazed plate heat exchanger and gasket-type plate heat exchanger. The brazed plate heat exchanger is difficult to clean when solid particles accumulate inside. A gasket-*

*type plate heat exchanger was selected because it uses a bolt-connected plate for easy disassembly during cleaning*".

4. What is targeted area for subglacial drilling? Arctica? Antarctica? The logistics and requirements for the system will vary depending on the potential drill site location.

The targeted area for subglacial bedrock drilling is Antarctica. In the revised manuscript, the word "polar region" is replaced by "Antarctica".

5. The environments of the domestic tests conducted in China and Antarctica differ greatly in terms of temperature and the nature of the drilling material (firn or ice vs sandstone). As a result, these tests cannot provide reliable evidence of the system's performance in polar regions. In addition, a lot of information about field tests, such as the rate of penetration, the volume of recovered chips, the diameter of the drilled borehole, etc. is not included.

It is true that the testing environments in China and Antarctica are quite different. However, it is difficult to create a similar testing environment as the Antarctica in China, so the domestic test is only used for checking the basic function of the drilling equipment. In the revised manuscript, we mentioned "*the testing environment in China is quite different with Antarctica and the domestic testing can only check the basic function of the air system*".

The following sentences are added to provide more information about testing results. "*The upper 42 m of the borehole was drilled by a 178 mm drill bit made by steel. Total 25 days were spent to deepen the hole from surface to 42 m because of many unexpected problems in the drill bit and air the system.*", "*The diameter of the borehole from 31m to 110 m was 102 mm and a production drilling rate was about 8-12m/day*". The volume of recovered chips was not measured in field test and was unknown.

More details are beyond the scope of the manuscript and can be found in our coming paper about the testing result in Antarctica.

6. English language is primitive and, in some places, awkward.

Language of the revised manuscript has been edited by native English speaker.

**Comments on the text:**

L22. "Polar regions, which mainly include Arctic and Antarctica, …": The term Polar Regions has a more precise definition.

"Polar Regions" was replaced by Antarctica in the revised manuscript. The explanation of the polar region at the beginning of the manuscript has been deleted.

L32, 33. "by USA", "USA drilled" – better US drillers or US scientists.

Changed to "US scientists"

L34. "China retrieved" – Chinese drillers.

Changed to "Chinese drillers  retrieved…"

L35. I'd suggest also to mention Chinese-Russian subglacial drilling project: Talalay PG, Leitchenkov G, Lipenkov V., Sun Y, Zhang N, Gong D, Liu Y, Li Y, Sun Y, Abdrakhmanov I, Vorobyev M, Khalimov D, Fan X, Salamatin A, Ekaykin AA, Bing Li B (2025). Rare ice-base temperature measurements in Antarctica reveal a cold base in contrast with predictions. Commun Earth Environ 6, 189. doi.org/10.1038/s43247-025-02127-1

The following sentences has been added to provide more information about Chinese attempt in subglacial bedrock drilling "*In 2018/2019 season, …. Five years later, Chinese-Russian subglacial drilling project recovered a subglacial bedrock core of 0.48 m beneath 541 m ice sheet (Talalay et al., 2025)*".

References

*Talalay, P. G., Leitchenkov, G., Lipenkov, V., Sun, Y., Zhang, N., Gong, D., Liu, Y., Li, Y., Sun, Y., Abdrakhmanov, I., Vorobyev, M., Khalimov, D., Fan, X., Salamatin, A., Ekaykin, A. A., Li, B.: Rare ice-base temperature measurements in Antarctica reveal a cold base in contrast with predictions. Commun. Earth. Environ., 6, 189, https://doi.org/10.1038/s43247-025-02127-1, 2025.*

L40. Truffer et al., 1999 used Longyear Super 38 drill rig.

The Longyear Super 38 drill rig was mentioned in revised manuscript.

L45. "it has a greater ability to deal with drilling accidents" – questionable statement.

The sentence was deleted.

L54-55. "it required ambient temperatures below ~10 °C to cool the compressed air to below 0°C" – not clear.

In the reference, it is mentioned that "*However, the temperature of the air leaving the aftercooler is still about 10 ℃, above ambient, so ambient temperatures must be below -10 ℃ or the air will melt the ice chips*."

The sentence was rewritten as "*...it required ambient temperatures below ~10 ℃, or the air will melt the ice chips*".

L62-63. "oil-based drilling liquid" – RAID used ESTISOL™ 140 that is not oil-based fluid.

The word "oil-based" was deleted.

L70-71. The issue of ice hydrofracturing is more complex than it may seem from a brief description in two sentences.

It is true that the ice hydrofracturing is complex. To address this, another sentence was added. "*At present, the occurrence condition of ice  hydrofracturing is still not clear and there is no effective way to prevent it from happening*"

L79. "the ice that was leftover" - ??

"the ice that was leftover" was changed to "the ice below".

L83-84. "newly designed" contradicts to "were improved"

The word "were improved" was deleted.

L91. What are "and so on"?

"and so on" was replaced by "size of ice chips and ice sheet temperature"

L96. "In theory, the compressed air should be, at least, cooled to below 0 ℃." Why? To prevent ice melting?

""to prevent ice from melting" was added after "below 0 ℃"

L97. "The dew point of compressed air was expected to be lower than -40 ℃" Why?

The following sentences were added to explain the reason why the dew point of compressed air was expected to be lower than -40 ℃. "*At the dew point of -40 ℃, the water content in the compressed air drops to 0.176 g·m⁻³, which is considered to be very dry*"

L98. "The air system is required to work at temperature of -30 ℃". It depends on potential drill sites. Where is targeted area for subglacial drilling?

The targeted drilling area of the MPDS has a distance less than 100 kilometers away from Antarctica coast. The following sentence was added. "*The targeted drilling area of the MPDS has a distance*

*less than 100 km away from Antarctic coast. In the targeted drilling area, the average atmosphere temperature is usually less than 30℃ (Wang and Hou, 2011). Consequently, the air system is also required to work at temperature of -30 ℃…"*

L113. "In time" – better, "during operation'.

Changed.

L126-127. The obvious statement. It's better to delete: "to monitor the flow rate, pressure, temperature, and dew-point of compressed air injected into a borehole".

The sentence was deleted.

L135. "because 10–20 % of air could be lost during the drying and cooling processes at the surface" – some robust estimations should be provided to support these numbers.

According to the manufacturer of the air system. About 8% of compressed air is lost to remove condensed water from the desiccant dyer. The freezing dryer can loss 0.5% of compressed air during water drainage. In addition, the air filters and air receivers also loss some compressed air during water drainage. In total, the lost air is more than 10%. The following sentences were added. "*During drying and cooling processes of compressed air at the surface, about 8% of compressed air is lost to remove condensed water from the desiccant dyer. The freezing dryer can loss approximately 0.5% of compressed air during water drainage. In addition, the air filters and air receivers also loss some*

*compressed air. In total, the lost compressed air could be more than 10%. In results, the selected compressor had a flow rate of 12 Nm³/min*".

L151. How much does the air temperature rise in adsorption tower?

According to the manufacturer of desiccant dyer, the air is usually heated to 180–220℃ to dry the moist desiccant. The original sentence was rewritten as "*The absorption tower is externally heated with power of 4.5 kW to heat the compressed air to 180-220℃. And then, the heated compressed air is used to dry the moist desiccant*".

L151-152. "it takes about 15–30 minutes to dry a tower". Is this time enough to dry air with the flow of 10 Nm3·min-1?

Our description made a misunderstanding. According to the manufacturer, 15–30 minutes are required to dry the moist desiccant in one tower. It doesn't mean that 15–30 minutes are required to dry compressed air. The sentence has been changed to "*Generally, it takes about 15–30 minutes to dry the moist desiccant in one absorption tower*".

L153-154. -10 C is also below zero. The concrete temperature range for cooling in the first stage needs to be determined.

In the first stage, the air can be cooled to the temperature of -10–0℃. The sentence has been changed to "*first, to a temperature of -10–0℃, and then to below -10 ℃*".

L160. "air filter precision"?

In table 1, the "filter precision" is replaced by "filtration rating", which is a more common description of air filter precision.

L164. "Left and right" are relative terms that depend on a person's point of view.

It is right. To clearly demonstrate the left and right parts of the air system, the following sentence was added "*In this paper, the left part refers to the half container close to the triangular bracket of the sled, while the right part refers to another half container*".

L198 "did not work" or did not test?

Should be "did not test". It has been changed in revised manuscript.

L199. Fig 8 is cited before Fig. 7.

Fig 8a has been deleted.

L210 and further. Was the temperature measured with precision of ± 0.01 C?

The precision of the temperature sensor used in the air system is ±1℃. In original manuscript, we used average value of measured data with precision of ±0.01℃ to show the testing results. In revised manuscript, the average value only shows the precision of ±1℃. It is worth to be mentioned that the dew-point hygrometer used in air cooler has a ±0.1℃.

L209-220. The description is not very clear. I suggest to modify text or convert it to a table.

[revised manuscript text omitted]

L258. It is not report. It is paper or article.

The sentence has been changed to "*The flow rate and pressure required for ice and subglacial bedrock drilling below 1000 m was roughly estimated according to the method of Alemany et al. (2021)*".

L257-258. Refereeing to Alemany et al. (2021) in order to choose flow rate and pressure required for ice and subglacial bedrock drilling is incorrect method because borehole depth, diameters of

drill pipes and bits are absolutely different. Authors should provide their own estimation of surface pump requirements.

We only use the method provided by Alemany et al. 2021, instead of choosing the flow rate and pressure according to their results. Our estimation is based on the type of drilling liquid, borehole depth, borehole diameter, and drill-rod size. The details of the estimation are as following.

*The flow rate and pressure of drilling liquid required in ice drilling and subglacial bedrock drilling can be calculated according to Alemany and others (2021). In the calculation, the penetration rate is also set as 100 m/h, while the concentration of ice chips is assumed to be 0.025. Kerosene JET-A1 is assumed to be drilling liquid. The drilling fluid are considered to be at the temperature of $-30$ °C, which has a density of 846 kg/m³ and a viscosity of $5.29 \times 10^{-6}$ m²/s (Talalay and Gundestrup, 2002). The safe factor of pressure is set as 1.5. In both ice drilling and subglacial bedrock drilling, the maximum drilling depth is considered to be 1000 m. It is worth to be mentioned that liquid reverse circulation is used in ice drilling while liquid normal drilling is used in subglacial bedrock drilling.*

*As shown in the following Fig. S1, the required flow rate keeps constant with increased drilling depth. In our case, the flow rate is about 76 L/min. However, the required liquid pressure during ice drilling increasing linearly with drilling depth increasing. In a 1000 m borehole, the required liquid pressure can be more than 1.0 MPa. The required flow rate and pressure in subglacial bedrock drilling has the same variation trends as in ice drilling. To obtain bedrock sample below 1000 m ice, the required flow rate is less than 70 L/min and the required pressure is about 1.6 MPa. To ensure the ability of the drilling-fluid pump, it is expected to have a rated flow rate of 100 L/min and maximum pressure of the 2 MPa.*

[Figure]

***Figure S1. Flow rate and pressure of drilling liquid required in ice drilling (a) and subglacial bedrock drilling (b)***

As mentioned above, the estimation of the flow rate and pressure needs the type of drilling liquid, borehole depth, borehole diameter, and drill-rod size. The detailed information will be provided in our coming paper about the whole MPDS, including the calculation process of flow rate and pressure. In the study, we focus on the design and test results of the drilling liquid circulation system, so the estimation is not contained in the manuscript.

L261. Why 2 MPa. Reference?

See the explanation above.

L261-263. A poor comparison. Both the RAID and SUBGLACIOR systems were designed to drill more than 3000 meters.

The comparison with RAID and SUBGLACIOR has been deleted.

L263. "The drilling fluid must be colder than -2 ℃ in polar regions" – Reference?

Theoretically, the drilling fluid must be colder than 0 ℃ in polar regions to prevent ice from melting. In our case, -2℃ is expected. The following sentence are added. "*The drilling fluid must be colder than 0 ℃.... In our case, the temperature of drilling liquid is expected to be lower than -2℃*".

L271-272. "The drilling liquid need not…" – sentence is not clear.

The sentence was changed to "*The drilling liquid don't need to be actively cooled…*".

L333. "Left and right" are relative terms that depend on a person's point of view.

The following sentence has been added in the revised manuscript. "*In this paper, the left part refers to the half container close to the triangular bracket of the sled, while the right part refers to another half container*".

L386-392. Language problems.

The sentences were reorganized as "*In the field, the flow rate of drilling liquid had to be regulated with a pressure-relief valve by operator. Besides, the circulation pump could not suck the drilling liquid from settling tank to vibration sieve because the liquid level in settling tank was lower than the circulation pump. The ice chips became very hard after being centrifuged. Removal of the hard ice chips from centrifuge drum was difficult and labor-intensive. In addition, there was no brake on the vertical centrifuge and much time was wasted in waiting for it to stop before cleaning it*".

L400-401. What does "vertical centrifuge … with brake and self-cleaning function" mean?

There is a type of vertical centrifuge, which has buit-in brake and can remove the hard ice chips without human. The sentence was changed to "*Fourth, a type of vertical centrifuge with built-in brake should be used and the vertical centrifuge should have the ability to remove the hard ice chips by itself, instead of operator*".

L406-407. Is this conclusion??

The name of the section was changed to "Experiences and lessons".

L 408-409. I disagree with the statement that "the design principle of the air system was proved to be feasible." The system requires significant modifications and additional testing.

The sentence was changed to "The testing results shows that the air system requires significant modifications and additional testing".

L412-413. "For atmospheric temperatures lower than -20 °C, a passive heat exchanger with a large air fan is strongly recommended for cooling compressed air." This system was neither designed nor tested. Therefore, how can the authors "strongly recommend" it?

The sentence was changed to "*For atmospheric temperatures lower than -20 °C, the feasibility of cooling compressed air by a passive heat exchanger with an air fan is worthy to be studied*".

L420-421. It seems that the separation system has not worked well. Therefore, the system needs significant modifications and further testing.

The sentence was changed to "*The vibration sieve and the vertical centrifuge required significant modifications to separate ice cuttings from the drilling liquid efficiently*".

**Comments on the figures:**

Figure 1. Schematic diagram looks a bit strange without any connections (lines) between subsystems.

The electricity line and the air/drilling liquid pipeline were added to Fig.1.

Figure 4. I suggest deleting one of the 3D models and installing all signs on only one of them.

If we deleted one of the 3D models, it can't install all signs on only one of them, so both the 3D models should be kept.

Figure 5. It is not informative - I suggest deleting it.

Deleted.

Figure 6. It is not informative - I suggest deleting it.

Deleted.

Figure 7. Commenting on the figure, it would be better to talk about ranges rather than average values.

In the revised manuscript, the data ranges are mentioned. However, the average value still be kept.

Figure 11. I suggest deleting one of the 3D models and installing all signs on only one of them.

If we deleted one of the 3D models, it can't install all signs on only one of them, so both the 3D models should be kept.

Figure 12. It is not informative - I suggest deleting it.

Deleted.

Figure 13. It is not informative - I suggest deleting it.

Deleted.

Figure 14. Commenting on the figure, it would be better to talk about ranges rather than average values.

In the revised manuscript, the data ranges are mentioned. The test results of the air system also were presented in ranges instead of average values.

---

## Author Comment (AC2)

**Reply to the comments of refree#2**

**Summary:** The paper details a promising drilling facility tailored for polar conditions. However, to enhance the overall description of such equipment, clearer articulation of the research gap and system advantages, and stronger connections between design choices and operational outcomes for both Air system and DFCS, would be welcome.

Thanks for your fruitful comments. The manuscript has been carefully revised according to your suggestion.

**Remarks and suggestions:**

- Abstract:

1. Suggest starting with a scientific/operational challenge (e.g., xxx current problem in sampling of subglacial bedrock in polar regions… this is why this type of MPDS is needed).

   The following sentences has been added at the beginning of abstract '*Liquid drilling is commonly utilized in sampling of subglacial bedrock in Antarctica. However, this drilling method has relatively low penetration rate compared with air drilling. Additionally, the drilling method may lead to hydraulic fracturing of ice borehole. In this study, A multi-process drilling system (MPDS) incorporated with different drilling methods…*'

2. The abstract is focusing heavily on the list of components, suggest highlighting the unique integration or operational principle that makes this MPDS different.

   The two sentences '*The air system comprised a compressor, receiver, freezing dryer, desiccant dryer and cooler. The DFCS comprised a refrigerating machine, heat exchanger, circulation pump, drilling-fluid pump, vibration sieve, vertical centrifuge, circulation tank, stirring tank, and settling tank*' were rewritten as '*The air system uses a compressor to generate compressed air at a flow rate of 10 Nm3·min-1 and maximum pressure of 1.5 MPa. The compressed air was then dried by a freezing dryer and a desiccant dryer to a dew point of -40°C. Before injected into the borehole, the compressed air was cooled to ≤-5 °C by an air cooler. The DFCS can pump drilling fluid to the borehole at a flow rate of 100 L·min-2 and maximum pressure of 2 MPa. The drilling liquid can be cooled to ≤-5 °C by a refrigerating machine and a heat exchanger within DFCS. The ice or rock cuttings are separated by a vibration sieve and a vertical centrifuge*'.

3. The narrative could be more sharply focused on key innovations and results. Including at least 2 or 3 key quantified performance metrics (for e.g., air system pressure, DFCS fluid temperature, maximum drilling depth achieved, or...), will help evaluate the advance.

Some key parameters of the two system have been presented in abstract as '*The air system uses a compressor to generate compressed air at a flow rate of 10 Nm3·min-1 and maximum pressure of 1.5 MPa. The compressed air was then dried by a freezing dryer and a desiccant dryer to a dew point of -40℃. Before injected into the borehole, the compressed air was cooled to ≤-5 ℃ by an air cooler. The DFCS can pump drilling fluid to the borehole at a flow rate of 100 L·min-2 and maximum pressure of 2 MPa. The drilling liquid can be cooled to ≤-5 ℃ by a refrigerating machine and a heat exchanger within DFCS. The ice or rock cuttings are separated by a vibration sieve and a vertical centrifuge*'.

- Introduction:

1. While the introduction reviews many past projects, it could more clearly state what existing systems lacked and how the MPDS, DFCS solve them. It would be better to add a clear, simple sentence or two just before introducing the MPDS that says exactly what was missing in previous work (e.g., insufficient fluid control in deep ice drilling or lack of integrated, transportable drilling systems, or … something else?). Currently, this research gap is implied but not explicitly stated.

The following sentences were added.

[revised manuscript text omitted]

3. The introduction concludes nicely. However, the justification for choosing a reverse circulation design could be expanded. Is this the first application of reverse circulation in subglacial or polar drilling? If so, why is this approach necessary? If not, what specific advantages does it offer over prior implementations?

Reverse circulation has been used in ice drilling in Antarctica. In past Antarctic drilling projects, reverse circulation is usually established in normal drill rod. In this case, drilling fluid is pumped through the clearance of drill rod and borehole wall, and then flows back to surface through the central passage of the drill rod. The drilling fluid continuously erodes the borehole wall. The MPDS uses double-wall drill rod as flow channel of compressed air or drilling liquid. Compressed air or drilling liquid is injected through the inner and outer tube of the double-wall drill rod and returned to surface through the central passage of the inner tube.

The following presents several advantages of reverse circulation with double-wall drill rod.

(1) During snow/firn drilling with compressed air, reverse circulation with double-wall drill rod can effectively prevent the leakage of compressed air into the surrounding snow;

(2) During ice drilling with drilling liquid, reverse circulation with double-wall drill rod can avoid the erosion of drilling liquid to the borehole wall. Further, it can prevent possible hydraulic fracturing of ice borehole wall.

The following sentence has been added in revised manuscript '*During drilling with reverse circulation, the compressed air or drilling liquid is injected through the inner and outer tube of the double-wall drill rod and returned to surface through the central passage of the inner tube*'.

'*During snow and firn drilling, compressed air with reverse circulation was used, which can effectively prevent the leakage of compressed air into the surrounding snow*'.

'*In this way, the erosion of drilling liquid to the borehole wall can be avoided. Further, it may help in preventing possible hydraulic fracturing of ice borehole wall*'.

- Air system:

1. Authors note that the freezing dryer and desiccant dryer failed in the field. This is important, but the cause analysis is too brief. Authors have mentioned the reason as 'groundwater presence', but the section could better emphasise the implications of these failures for long-term operation in Antarctic conditions. Also, how severe was the downtime caused by repeated ice plug formation in terms of drilling efficiency?

The reason of the two dryer's failure in field is reanalyzed as following '*It was found that a lot of the condensate was not discharged from the air receiver through the drain valve (Fig. 9a). We suspect it is one of the reasons why the freezing dryer and the desiccant dryer could not dry the compressed air to desired dew-point. Additionally, lot of the condensate was also found in the freezing dryer and the air filters, which share the drainage channel. It is suspected that the drainage channel was blocked somewhere or the drain valve on the freezing dryer and the air filters stopped to work. Manufacturing defects of the freezing dryer and the desiccant dryer were also suspected. Due to limited working time in Antarctic field, the air system was not carefully checked and the specific reasons for the failure of the freezing dryer and the desiccant dryer remains unclea.*'.

The 'groundwater presence' is not the reason for the failure of the freezing dryer and the desiccant dryer in Antarctica field. In domestic test, the MPDS was used for drilling underground soil and rock and the groundwater was found in the borehole. Consequently, the freezing dryer, desiccant dryer, and cooler were determined not to be used in the testing.

The implications of these failures for long-term operation in Antarctic conditions were emphasized with following sentences '*Overall, long-term drilling with compressed air in Antarctic without freezing dryer and the desiccant dryer is unrealistic*'.

The downtime caused by repeated ice plug formation has been evaluated using the following sentences '*In this case, the condensate froze into an ice plug in the air-cooler conduit leading to a rapid rise in the outlet pressure of the air cooler. In common case, after 15–20 minutes, the pressure increased from 0.7 MPa to 1.3 MPa, which was close to the limited maximum pressure of air cooler. In the field, the ice plug was melted by hot compressed air after shutting off the air cooler. In this situation, the drilling work had to be stopped until the pressure dropped to normal values. This process took approximately 5–8 minutes. In addition, another 3–5 minutes were required to restart the cooler*'.

To make it more clear, other sentences has been added in revised manuscript '*Frequent ice plug formation in the air-cooler conduit significantly hinders continuous ice drilling operations, drastically reducing drilling efficiency. In some cases, drilling must be interrupted up to three times within a single run while waiting for the air system to be restored to operational readiness*'.

2. How the MPDS air system's cooling/drying capacity exceeds or differs from RAM, Winkie drill setups, etc.? Is it due to component selection, integration, or operational setup?

Winkie drill can only drill with liquid, so it has no air system.

Air treatment subassembly of RAM was modified to provide two stages of cooling and water separation. In addition, a coalescing filter was included as a means to remove any remaining water vapor. According to Gibson et.al (2021), compressed air can be cooled to temperature of 0°C to −10°C by adjusting fan speed and baffles on the aftercoolers. The cooling capacity of RAM is comparable to that of MPDS. The drying capacity of the RAM is not shown in public literature and still unknown.

The following sentences have been added in the revised manuscript '*In general, the air cooler can cool the compressed air effectively and its cooling capacity is comparable to that of RAM-2 drill (Gibson et. al, 2021)*'.

The table comparing domestic test and Antarctic field test is shown in the following table. However, we think it is not necessary to present the table in the manuscript, because all the testing data has been shown in figure 8 and in text. Additionally, the data can also be found in supplement material Table S1.

**Testing results of the air system**

| Property of compressed air | Domestic test | Field test in Antarctica |
|---|---|---|
| Flow rate/Nm$^3\cdot$min$^{-1}$ | 10.9 | 9.3 – 12.4 |
| Pressure/MPa | 1.5 | 0.7 – 1 |
| Temperature after compressing/°C | 95 – 97 | 76 – 89 |
| Temperature at the inlet of freezing dryer/°C | / | 24 – 44 |
| Temperature at the outlet of freezing dryer/°C | / | ~35 |
| Temperature after first cooler (first cooler worked)/°C | -4.5 | -17.4 – -7.9 |
| Temperature after second cooler (first cooler worked)/°C | / | -12.4 – -4.9 |
| Temperature after first cooler (second cooler worked)/°C | / | 17.6 – 41.7 |
| Temperature after second cooler (second cooler worked)/°C | / | -15.9 – -5.1 |
| Temperature after second cooler (both coolers worked)/°C | -10.8 | / |
| Dew point at the inlet of the cooler/°C | / | -11.3 – 19.9 |
| Dew point at the outlet of the cooler/°C | -48–-52 | -21.2 – 4.6 |

- DFCS:

1. The section lists components but does not explicitly connect each one to the drilling challenges described in the introduction (e.g., how each device addresses heat, humidity, or borehole icing).

In section 2.1.1 of the revised manuscript, the following sentences were added.

'*Compared with air-cooling methods used in the past, using refrigerant to cool compressed air ensures a consistently sub-zero temperature (<0°C) regardless of external atmospheric variations.*'

'*Overall, the air system employs a two-stage dehumidification process, offering greater reliability than conventional air-drying methods used in polar regions*'.

In section 2.2.2 of the revised manuscript, the following sentences were added.

'*In this way, the drilling fluid can be actively cooled to a low temperature, preventing the melting of the ice-borehole wall—a phenomenon observed in other subglacial bedrock drilling projects*'.

'*Comparing with the heated melter tank used by RAID and ASIG, the vertical centrifuge is more efficient and energy-saving (Kuhl et al., 2021; Goodge et al., 2021)*'.

2. Similar to Air System, this section also would benefit from a table comparing domestic test vs. Antarctic field performances.

The table comparing domestic test and Antarctic field test is shown in the following table. However, we think it is not necessary to present the table in the manuscript, because all the testing data has been shown in figure 10 and in text. Additionally, the data can also be found in supplement material Table S2.

**Testing results of the DFCS**

| Property of drilling liquid | Domestic test | Field test in Antarctica |
|---|---|---|
| Flow rate/L·min$^{-1}$ | 110 | 62–96 |
| Pressure/MPa | / | 0.15–0.29 |
| Temperature/°C | 2 m$^3$ water could be cooled from 24.8 to 7 °C in 50 minutes | -15.5–-4.5 |

3. The list in 3.4.2 is useful, but each suggestion could be strengthened by linking why it is needed to the actual observed field problem (e.g., "flow rate should be adjustable… because … Connect this back to the results"). Where possible, quantify desired capacities (e.g., "small stirrer" how small? Specify dimensions).

Some sentences have been added to strength the link between the suggestion and the observed field problem. The revised sentences are shown as following '*First, the flow rate of the drilling-fluid pump should be adjustable by variable-frequency drive for easy operation. Otherwise, the drilling-fluid pump has to be manually operated and can easily lead to mismatch of flow rate with drilling requirement. Second, the pump with suction capacity should be used as circulation pump, such as self-priming pumps and submersible pumps. Third, a high-capacity vibration sieve with double layer of screens is suggested to improve its processing capacity of drilling fluid. Fourth, a type of vertical centrifuge with built-in brake should be used and the vertical centrifuge should have the ability to*

*remove the hard ice chips by itself, instead of operator, to reduce the intensity of manual labour. Additionally, small stirrer with size less should be added on the circulation tank, and a filter screen made by steel wire are strongly suggested to add between the circulation pump and the heat exchanger to prevent the settled clays block the flow channel'.*